# *Chlamydia trachomatis* Cross-Serovar Protection during Experimental Lung Reinfection in Mice

**DOI:** 10.3390/vaccines9080871

**Published:** 2021-08-06

**Authors:** Christian Lanfermann, Martin Kohn, Robert Laudeley, Claudia Rheinheimer, Andreas Klos

**Affiliations:** Institute of Medical Microbiology and Hospital Epidemiology, Medical School Hannover, 30625 Hannover, Germany; Lanfermann.Christian@mh-hannover.de (C.L.); Kohn.Martin@mh-hannover.de (M.K.); Laudeley.Robert@mh-hannover.de (R.L.); Rheinheimer.Claudia@mh-hannover.de (C.R.)

**Keywords:** *Chlamydia trachomatis*, reinfection, cross-serovar, protection, vaccination, lung, mouse, animal model

## Abstract

*Chlamydia trachomatis* causes most bacterial sexually transmitted diseases worldwide. Different major outer membrane proteins (MOMPs) define various serovars of this intracellular pathogen: In women, D to L3 can cause urethritis, cervicitis, salpingitis, and oophoritis, and, thus, infertility. Protective immunity might be serovar-specific since chlamydial infection does not appear to induce an effective acquired immunity and reinfections occur. A better understanding of induced cross-serovar protection is essential for the selection of suitable antigens in vaccine development. In our mouse lung infection screening model, we evaluated the urogenital serovars D, E, and L2 in this regard. Seven weeks after primary infection or mock-infection, respectively, mice were infected a second time with the identical or one of the other serovars. Body weight and clinical score were monitored for 7 days. Near the peak of the second lung infection, bacterial load, myeloperoxidase, IFN-γ, and TNF-α in lung homogenate, as well as chlamydia-specific IgG levels in blood were determined. Surprisingly, compared with mice that were infected then for the first time, almost independent of the serovar combination used, all acquired parameters of disease were similarly diminished. Our reinfection study suggests that efficient cross-serovar protection could be achieved by a vaccine combining chlamydial antigens that do not include nonconserved MOMP regions.

## 1. Introduction

Chlamydia are Gram-negative, obligate intracellular bacteria with a unique productive cycle alternating between two morphological and metabolically distinct forms. The extracellular, infectious form—the elementary body (EB)—possesses a rigid cell wall and almost no metabolism. It invades preferably mucosal epithelial cells [1]. Inside the host cell, chlamydia survive in a vacuole-like compartment. In this inclusion, they acquire nutrients, interfere with host signaling pathways and cell functions, and transform into their metabolically active and propagating second form—the reticulate body (RB). After a few days, a new generation of EBs is released from the cell, either by cell lysis or by extrusion. Currently, 17 species of chlamydia have been identified. Chlamydia has a wide host range and tissue specificity. In humans, some species can cause infections of the urogenital and respiratory tract and the eye. In rare cases, gastrointestinal infections have also been reported [2].

The species *Chlamydia trachomatis* (*C.tr.*) consists of 19 known serovars. D to K represent the genital tract biovar, and L1 to L3 the lymphogranuloma venereum (LGV) biovar [1,3]. These serovars mainly infect the urogenital tract, but they can also cause conjunctivitis. With an estimated prevalence of 4.2% in men and women between 14 and 49 years of age worldwide and with 124 million new yearly cases (WHO, 2016), *C.tr.* is the most frequent bacterial cause of a sexually transmitted disease (STD). In particular, in women, acute genital *C.tr.* infections remain largely asymptomatic. Unrecognized, they can cause tissue damage of the fallopian tubes, ovaries or uterus. Potential sequelae of an untreated chlamydial genital infection include pelvic inflammatory disease and ectopic pregnancy. It has been estimated that in approximately 1 out of 200 cases, *C.tr.* infection results in permanent tubal infertility [4]. During birth, infected pregnant women can transmit *C.tr.* to the infant, causing difficult-to-treat newborn pneumonia and conjunctivitis. Azithromycin, doxycycline or quinolones are effective against this intracellular pathogen. Most likely, partially due to an inefficient immune response, partially due to the inability of these antibiotics to completely eradicate bacteria in some patients, relapses or reinfections—finally leading to severe pathology—are common. Therapeutic failure is estimated to be 10% [5]. Furthermore, the strategy of several developed countries to decrease infection rates by providing the opportunity for an annual PCR screening for all young women paid by health insurance has remained ineffective. Only screening for *C.tr.*-DNA during pregnancy has been successful in industrialized countries [6]. 

Reactive arthritis is another secondary complication of urogenital *C.tr.* infections, mainly in men with a specific genetic background. In this disease, persistent viable *C.tr.* with reduced metabolism are detectable in the synovia of the affected joints for months to years [7]. A potential link between *C.tr.* infections and urogenital tumors is also discussed [8].

Ocular infections caused by the trachoma biovar of *C.tr.*, which include serovars A to C, remain a constant serious health problem in over 40, mainly developing countries (WHO GHO 2019 [9]). This severe eye infection is responsible for visual impairment, including blindness, of 1.8 million people. In 1993, the WHO adopted the SAFE Strategy (Surgery, Antibiotics, Facial cleanliness, Environmental improvement) to eradicate trachoma by the year 2020. However, it was found that this goal could not be achieved with such measures alone (WHO [10]).

In summary, *C.tr.* remains an important health problem that might best be targeted by a vaccine. As *C.tr.* has the ability to infect the same person repetitively (reviewed by Phillips et al. [11]), the development of a cross-serovar-effective, prophylactic vaccine against *C.tr.* is especially important.

The chlamydial outer membrane consists of a highly diverse and complex arsenal of many different proteins that aid the bacteria during cell adhesion, invasion or inclusion-forming. The *C.tr.* serovars are mainly differentiated by the composition of the four variable domains of their major outer membrane protein (MOMP) [12,13,14], which is encoded by the ompA gene. While large parts of the chlamydial genome are remarkably conserved, these sequences vary to a larger degree between serovars, enabling their identification. Moreover, MOMP induces *C.tr.* serovar or at least serogroup-specific responses, as reviewed by De La Maza et al. [1]. Therefore, successful MOMP-based subunit vaccines represent several serovars. The best example for this is a vaccine combining the variable domain VD4 and the surrounding constant immunogenic regions of MOMP molecules from several frequent serovars, which has already passed a phase-1 clinical trial [15,16]. Yet, after population-wide vaccination with parts of MOMP, escape mutants might emerge since the ompA gene can mutate to a certain degree without loss of function, as demonstrated by existing various *C.tr.* genotypes and serovars [17] or the hybrid *C.tr.* genomic mosaic L2b/D-Da strain, causing an outbreak of LGV [18]. Hence, other chlamydial antigens are still a potential suitable alternative (or auxiliary component) for vaccine development.

Next to MOMP, several chlamydial surface proteins have been proposed to act as adhesins, including several glycans and the protein family of polymorphic membrane proteins (Pmps) [19,20,21]. Both MOMP and Pmps are abundant on the chlamydial surface. However, in contrast to the MOMP variable structure between serovars, the nine Pmps (A–I) of *C.tr.* are highly conserved [22]. Hence, they are an example for a potential valuable alternative asset of antigens in the development of a cross-serovar protective vaccine. This has been demonstrated by several studies of others [23,24,25] and, most recently, our research group [26]. Thus, it is important to know how far induced protection by the immune system, independent of the serovar-specific MOMP regions, is effective. 

In contrast to humans, where usually only the lung of newborns is affected, adult mice develop pneumonia within 1 week after experimental i.n. *C.tr.* infection [26,27]. Of course, urogenital mouse models with their observation period of weeks to months after challenge infection will be better suited for studies confirming, for example, the effect of a vaccine against *C.tr.* in the intended context [26,27,28,29]. However, for screening purposes or a proof of principle, the mouse *C.tr.* lung infection model has many advantages. Rapidly developing pneumonia can be monitored exactly by several hallmarks of infection and animal well-being, including body weight and clinical score, bacterial load, the granulocyte marker myeloperoxidase (MPO), and levels of key cytokines in the infected organ [26,27]. 

In this study, we compared cross-serovar-specific with serovar-specific induced protection in a lung reinfection model with C57BL/6J mice applying three different serovars of the human pathogen *C.tr.* The protection elicited by the primary infection was characterized after a secondary infection by monitoring the aforementioned infection hallmarks, as well as the induced chlamydia-specific IgG responses. Serovars L2, D, and E of serogroup B were selected since they belong to the most frequent LGV or non-LGV serovars, respectively [3,30,31,32]. The results of this comparison provide information about the induced maximal achievable general and the “MOMP-independent” protection, and thus, about the potential of a future vaccine based on chlamydial antigens that do not include nonconserved MOMP regions. 

## 2. Materials and Methods

### 2.1. Chlamydial Culture

Three *C.tr.* serovars (*C.tr.* D UW3/CX, ATCC: VR-885; E provided by Hegemann, Düsseldorf; and L2 LGV II 434, ATCC: VR-902B) were cultured and propagated in baby hamster kidney 21 (BHK-21, ATCC^®^CCL-10) for the mouse lung infection experiments or in HeLa-T cells (Henrietta Lacks cervix carcinoma cell line, gifted from Heilbronn, FU Berlin) for IFU determination in cell culture. EBs were centrifuged (55 min; 35 °C; 2000× *g*) on a cell monolayer in Panserin 401 medium (Cytogen, Berlin) with 1 mg/mL cycloheximide (Sigma-Aldrich, St. Louis, MO, USA) for infection. After incubation at 37 °C and 5% CO_2_ atmosphere for 2 days, cells were detached and destroyed manually with a cell scraper and cell debris were removed by centrifugation (15 min; 500× *g*). The supernatant was centrifuged again (1 h; 4 °C; 22,000× *g*); EBs were washed with a transport medium (1× PBS containing 2% FCS, 6.86% sucrose, 0.04 mg/mL gentamicin, and 0.002% phenol red) and centrifuged one more time. The inclusion-forming units (IFU) of all preparations were measured and quantified by titration on HeLa-T cells and flow cytometric determination [26] and tested by PCR to exclude mycoplasma contamination. Negative (mock) controls were proceeded identically in BHK-21 cells while leaving out chlamydia during infection.

### 2.2. Animal Experiments and Clinical Scoring

As depicted in Figure 1, 7-week-old female C57BL/6J mice were treated intraperitoneally (i.p.) with 2.5 mg medroxyprogesterone acetate (MPA) in 200 µL, 0.9% NaCl to synchronize their estrous cycle. The age difference between individual mice of the experimental groups was no more than 5 days. On day 7 of the experiment, the animals received a primary intranasal (i.n.) infection with *C.tr.* The infected mice had to exhibit a severe, but still tolerable course of lung disease. In addition, at the time of second infection, the mice are twice as old as in the primary infection, and, according to titration experiments, usually more sensitive to chlamydia. To induce a similar severity of the resulting lung disease and ensure animal survival in all relevant groups, different IFU amounts of the serovars had to be used (see [26]). To avoid cross-infection, mice were housed according to their serovar, but were otherwise handled in parallel. For the i.n. infections, mice were anesthetized by i.p. injection of 0.1 mL anesthetic solution (100 mg/kg BW Anesketin, 4 mg/kg Rompun in 0.9% NaCl) per 10 g body weight. In the primary infection, *C.tr.* E (1.3 × 10^6^ IFU), D (6 × 10^6^ IFU) or L2 (8 × 10^5^ IFU) were applied and the animals were monitored daily for the following 2 weeks. At days 28 and 35, previously infected mice were given azithromycin (1.6 mg/20 g body weight) per oral gavage to eliminate any remaining bacteria [27]. The hormonal cycle of the mice was synchronized again on day 49 with MPA and, on day 56, they were infected a second time with either *C.tr.* E (1.3 × 10^6^ IFU), D (2 × 10^6^ IFU) or L2 (4 × 10^5^ IFU) with an observation period of additional 7 days. Control animals received mock material on either day 7, day 56 or both. Seven days after the second *C.tr.* infection, all mice were sacrificed painlessly and blood and lungs were harvested for analysis. In the lung homogenate, bacterial load (IFU), the levels of granulocyte marker myeloperoxidase (MPO) and of IFN-γ and TNF-α were measured. During the 7-day infection period, mice were closely monitored under humane endpoint-defining criteria. The body weight and clinical score (fur quality, body posture, locomotion, breathing, attention/curiosity, hydration, secretion from eyes/nose) were surveilled and determined daily [26]. The results shown usually combined data obtained from two to three staggered but otherwise identical experiments.

### 2.3. Determination of Bacterial Load in Mouse Lung with Flow Cytometry

The bacterial load was determined in lung homogenates from the right lung lobes, collected on day 7 of secondary infection (day 63 in Figure 1). The amount of vital chlamydia as inclusion-forming units (IFU) was determined in 24 h infected HeLa-T cells by staining for intracellular chlamydia with Pathfinder^®^ (Fluorescein-conjugated murine monoclonal antibody to chlamydial LPS; 0.1% Evans Blue; Bio-Rad) and flow cytometry, as described elsewhere [26]. 

### 2.4. Quantification of Cytokine and Myeloperoxidase Levels

On day 7 of secondary infection, the concentration of the cytokines IFN-γ, TNF-α, and of the granulocyte marker MPO, was measured in mouse lung homogenate by the enzyme-linked immunosorbent assay (ELISA) according to the manufacturer’s protocol (TNF-α—ELISA MAX™ Deluxe Set Mouse TNF-α, BioLegend, 430904; IFN-γ—ELISA MAX™ Deluxe Set Mouse IFN-γ, BioLegend, 430804; MPO—MPO, Mouse, ELISA kit, Hycult Biotech, HK210-02). Absorbance was measured at a wavelength of 450/540 nm (Synergy HTC Multi-Mode Reader Biotec^®^ plate reader).

### 2.5. Determination of Chlamydia-Specific IgG by ELISA

The level of specific IgG was determined by ELISA (modified from a recently described IgG ELISA using the homogenate of *C. psittaci* as the antigen [33,34]) in mouse serum harvested on day 7 of secondary infection. Polystyrene microtiter plates were precoated at 4 °C for 16 h with 100 µL of *C.tr.* serovars D, E or L2 homogenate as antigen (2.5 μg/mL per well in PBS, as confirmed by the BCA assay (Thermo Fisher, no. 23225) and indicated in the graphs). The chlamydia-free cell homogenate served as mock control. Nonspecific binding-sites were blocked by PBS containing 1% BSA and 5% sucrose. Sera obtained from D/D-, E/E- or L2/L2-infected mice, respectively, and those of control animals, were pooled within their respective groups and analyzed by serial dilution to determine their chlamydia-specific IgG antibodies. Absorbance was measured at 450/540 nm (Synergy HTC Multi-Mode Reader Biotec^®^ plate reader).

### 2.6. Statistics and Group Sizes

For statistical analysis, the conducted experiments were tested as follows: For the comparison of body weight, two-way ANOVA with Bonferroni’s post-test were utilized. The clinical score was analyzed using the Kruskal–Wallis test with Dunn’s multiple comparison post-test (≥3 groups) or the Mann–Whitney *t*-test (two groups). For chlamydial load in lung, IFN-γ, TNF-α, and MPO, one-way ANOVA with Bonferroni’s multiple comparison were used. For comparison of chlamydia-specific IgG in the serially diluted pool serum, two-way ANOVA with Bonferroni’s post-test were applied to the mean of the OD-values of the serial dilution. Except for body weight, clinical score, and the aforementioned OD-values, in most cases, the logarithmic transformation of parametric data was performed to achieve Gaussian distribution. Depicted *–*** indicate statistical significances (*p* < 0.05, <0.01, and <0.001) between the linked groups or if not indicated otherwise between the respective group and all other groups in the experiment. Additionally, in the figure depicting the IgG responses, using the identical *p*-values, #–### is used as described in the corresponding figure legend. All differences were considered significant at *p* < 0.05 or lower.

At the beginning of the experiments, the number of animals per group was usually *n* = 8–10, with the exception of *n* = 6 for the *C.tr.*L2/L2 primary and secondary infection and *n* = 4 for the Mock/Mock group. One animal from the Mock/D and two from the Mock/E group reached the humane endpoint prematurely and had to be removed from the experiment before the final analysis. For practical reasons, not all surviving animals could be analyzed regarding all parameters. In this case, samples collected for analysis at day 7 p.i. were chosen randomly and blinded. The resulting number of samples per group is indicated in the respective figure legend.

All statistics were calculated using the GraphPad Prism software, version 5.3 for Microsoft Windows (GraphPad Software, San Diego, California USA, www.graphpad.com).

## 3. Results

As a proof of principle, to elucidate how far induced cross-serovar and serovar-specific protection differ, strains representing three different genital *C.tr.* serovars were investigated in a mouse lung reinfection study. With the nine resulting serovar combinations (plus the corresponding mock-infected controls), determination of infection-induced protection was performed in hindsight of a potential prophylactic vaccine that does not contain variable MOMP domains.

### 3.1. Previous i.n. Infections with C. trachomatis Serovars D, E, and L2 Improve Body Weight and Clinical Score during a Later Cross-Serovar Infection

Following the protocol for primary and secondary infection (Figure 1), female C57BL/6J mice were infected with either *C.tr.* serovars D, E or L2 or mock material (as negative control) at two, 7-week-apart time points. This resulted in the following *C.tr.* serovar combinations: Mock/Mock, Mock/D, Mock/E, Mock/L2, D/D, D/E, D/L2, E/D, E/E, E/L2, L2/D, L2/E, L2/L2. During the 7 days after secondary infection, mice of the 13 groups were monitored daily to assess body weight (Figure 2(a1–a3)) and clinical score (Figure 2(b1–b3)). While Mock/Mock control mice showed no signs of disease, mice that encountered *C.tr.* at the later time-point for the first time (e.g., Mock/D, Mock/E, and Mock/L2) showed profound losses in body weight and high clinical scores during the course of disease (day 1–7). In contrast, animals that had already been infected with any of the three *C.tr.* serovars once before showed comparatively less weight loss (day 3 or 4 –7) and improved the clinical score (day 3–7), demonstrating infection-induced protection between all serovars. Interestingly, there were only slight deviations in efficacy between serovar-specific and cross-serovar achieved protection.

### 3.2. Initial i.n. Infections with C. trachomatis Serovars D, E, and L2 Lead to Similar Improved Bacterial Clearance after Secondary Infection with the Same or Other Serovars

After initial primary i.n. infection with either one of the three different *C.tr.* serovar strains (or mock material) and secondary infection, mice were painlessly killed on day 7 and the lung homogenate was harvested. When compared to the control group that encountered the corresponding *C.tr.* serovar only at the time of secondary infection (e.g., Mock/E), the amount of viable, infectious chlamydia in the lung homogenate of mice that had been infected with either serovar 7 weeks prior was more than 100× lower in serovars D and E secondary infection (Figure 3, left and middle graph), and 10× lower in secondary infection with L2 (Figure 3, right graph), respectively. While the difference between the Mock/serovar “X” and double infection groups in the bacterial load were always pronounced, the three serovars showed an almost identical potential for infection-induced protection between each other.

### 3.3. Myeloperoxidase Levels in Lung Homogenate Are Similarly Reduced during C. trachomatis D, E, and L2 Secondary Infection in Mice Previously Infected with Serovar E

Next, we measured levels of the granulocyte marker MPO at day 7 after secondary infection in the lung homogenates of mice that had been infected initially with *C.tr.* E or given mock material (Figure 4). Independent of the serovar used for secondary infection (D, E or L2), we found reduced levels of MPO—indicating reduced tissue inflammation—in mice that had overcome a previous *C.tr.* E infection. Concentrations of the granulocyte marker were then only slightly above the detection limit (and the level found in noninfected mock–control mice). For practical reasons, and due to the similar results obtained in the determination of the bacterial load, we restricted our analysis to these combinations of *C.tr.* serovars.

### 3.4. Levels of Key Cytokines IFN-γ and TNF-α Are Decreased to a Similar Extent during Homologous or Heterologous Secondary Infection with C. trachomatis D, E or L2

At day 7 of secondary infection, the cytokines IFN-γ (Figure 5(a1–a3)) and TNF-α (Figure 5(b1–b3)) were measured in the lung homogenate of mice for all possible serovar combinations. These cytokines, which are important components of the innate immune response against intracellular pathogens, are elevated in *C.tr.*-infected lungs [27]. When compared to the Mock/”X” control groups that encountered *C.tr.* for the first time 7 d before sacrifice, the IFN-γ and TNF-α levels were significantly decreased in all groups of animals that had already overcome a previous infection with either serovar 49 days before the secondary infection. The reduction of the cytokine levels determined after the secondary infection with serovars D, E or L2 was again very similar between the serovars used during the primary infection, indicating similar cross-serovar protective capabilities.

### 3.5. Antibodies Raised by Repetitive C. trachomatis Serovars D, E or L2 Lung Infection Almost Do Not Distinguish between Crude Antigen Preparations of the Three Serovars

To assess potential serovar-dependent differences in the induced circulating chlamydia-specific IgG antibodies, ELISAs were performed with serum samples obtained after homologous primary and secondary infection (Figure 6). Pooled serum from mock-infected control mice showed no measurable IgG response to the antigens of all three serovars, except in its lowest dilution (1:100), which is most likely due to unspecific binding. Pool sera from D/D-, E/E-, and L2/L2-infected mice tested on plates coated with chlamydia-free mock-material as the negative control did not lead to any IgG signal (OD < 0.1, data not shown). Specific IgG binding of the three sera from D/D-, E/E-, and L2/L2-infected mice to the crude antigen prepared from *C.tr.* serovar L2 was almost identical.

There were only minor differences in IgG binding to serovars D or E antigen preparations. If at all, pool serum from L2/L2-infected mice showed slightly less binding towards the *C.tr.* D and E antigens. We analyzed only pool sera obtained after reinfection with the identical serovar since orientating preliminary experiments had indicated higher sensitivity (data not shown). Indeed, all three pool sera were positive against all three antigens up to a dilution of 1:100,000 and higher.

## 4. Discussion

“To date, the MOMP has emerged as the most suitable substitute for whole cell targets” (as reviewed in [11]). Thus, MOMP plays a prominent role in the development of a *C.tr.* vaccine [1,35]. Yet, in addition to its only serovar- or serogroup-specific-induced protection and the potential risk of the selection of immune escape mutants, recombinant rMOMP seems to be less protective than native nMOMP purified from chlamydia, at least as shown for the very closely related species *C. muridarum* [15,36,37]. This further restricts its value as an antigen. Hence, there are several good reasons to consider alternative or additional non-MOMP subunit antigens in a *C.tr.* vaccine or even an attenuated isolate, such as the plasmid-deficient derivative of a *C.tr.* D strain that has been shown to be protective against genital infections of Rhesus macaques [38]. However, former studies with live or inactivated EBs of *C.tr.* led to the conclusion that the protection achieved is possibly only serovar- or serogroup-specific (as reviewed [38]). Based on these sources, it is difficult to obtain a clear, unambiguous picture of achievable, vaccine-induced, cross-serovar protection. Yet, this is a critical issue for *C.tr.* vaccine development.

Hence, in this study, we aimed to clarify this point as a proof of principle by performing systematic cross-infection experiments with three *C.tr.* serovars that cause frequently urogenital infection with its feared sequelae in humans, especially women. Female C57BL/6J mice were repetitively i.n. infected with *C.tr.* strains representing serovars D, E, and L2. For this study, the mouse lung (re)infection model seemed to be optimally suited [27]. In this model, close to the peak of infection of naive mice, achieved protection can be exactly quantified by various laboratory parameters, in particular, with bacterial load in the lung as the most important one. Moreover, daily determined body weight and clinical score (and humane endpoint defined survival) constantly depict the intended overall severe but still tolerable course of disease.

The aforementioned advantages of the lung infection model were also important during the indispensable establishing and optimizing serovar titration experiments in 8- and 15-week-old animals [26]. We wanted to achieve maximal sensitivity in detection of protection induced by primary infection. Therefore, the specific immune response had to be stimulated by the first infection as much as possible within the limit of tolerable distress of animal health. In addition, to be able to detect potential decreases in any of the parameters determined after the second i.n. application of the *C.tr.* serovars, infection-dependent changes of naïve control mice had to be rather high. The two daily determinable surrogate markers of the severity of infection and disease permitted selection of the necessary (relatively high) amount of chlamydia. Furthermore, reinfection or vaccination-challenge experiments are intrinsically of longer duration. In this study, e.g., the bacteria were applied at two 7-week-apart time points. Yet, with the relatively short required observation period of approximately 1 week after secondary infection, the lung model still enabled a relatively fast and thorough screening and quantitative comparative analysis of the numerous combinations of the three selected serovars and mock controls.

We have already used the i.n. *C.tr.* infection model for identification of a member of a new class of antibiotics [27,39], and, most recently, for successful characterization of a promising new multi-subunit vaccine against *C.tr.* [26]. Nevertheless, when considering the most important *C.tr.* diseases in humans and the final aims of therapy or vaccination, the limitations of this animal model must also be considered. Consequently, the mouse lung infection model can only complement, for certain biological or medical aspects, the valuable, but also not in every aspect ideal, urogenital animal models.

Our results obtained after *C.tr.* infection in the serovar combinations D/D, E/E, and L2/L2 show that the course of a second infection with the identical serovar is strongly ameliorated. Yet, the secondary infection was not blocked in its entirety, as demonstrated by transient weight loss and the initial increase in clinical score. In fact, the course of disease of mice during the first 1 or 2 days after the second infection was almost indistinguishable from that of control mice that had received mock material rather than chlamydia during the primary infection. The health of reinfected mice only started to improve from day 3 onwards. This brief delay argues against a prominent role of circulating neutralizing antibodies in emerging induced acquired protection—in particular, those of longer half-life, i.e., *C.tr.* specific IgG.

The kinetics of the course of disease in naïve mice seems to differ slightly between the serovars. While animals infected with *C.tr.* serovars D and E were still at the peak of disease at the end of the observation period, mice infected with *C.tr.* serovar L2 already showed hints of the beginning of recovery on days 6 and 7. This might explain why less chlamydial EBs or IFN-γ and TNF-α, could be detected in the lung homogenates of mice with initial mock infection 1 week after their secondary infection with serovar L2, compared with D or E.

Intriguingly, initial infection with any of the three serovars provided (with only minimal differences) comparable protection to a second infection, again, with any of the serovars, strongly indicating induced cross-serovar protection from overcome infections between *C.tr.* D, E, and L2. Although slightly delayed, infection-induced protection was rather effective, e.g., with bacterial loads, which were smaller by 1 to 2.5 log_10_ levels compared to control mice. Moreover, according to all other registered parameters, mice recovered almost completely within a few days after secondary infection if their immune system had encountered any serovar of *C.tr.* in a primary infection 7 weeks before.

Viable chlamydia could actively influence and partially suppress the immune response in the primary infection by a translocated effector. Hence, infection-induced protection does not only depend on the quantity, quality or “specificity” of the antigens expressed by a pathogen. Nevertheless, the similarly diminished course and severity of disease in all combinations of the three serovars still support a high degree of induced cross-serovar protection. Moreover, the similar observed induced protection is obviously independent of the differences in their MOMPs.

Based on the promising findings of this reinfection study, we started and recently published the results of a mouse vaccination study with a new five-component subunit vaccine (5cVAC). It combines PmpA, PmpD, PmpG, PmpH, and Ctad1, five EB surface antigens of *C.tr.* serovar E in their recombinant, purified form, with c-di-AMP as an adjuvant [26]. The first of three i.n. applications of this vaccine and the lung challenge infection with various serovars of *C.tr.* were performed 7 weeks apart, i.e., similar to the interval between primary and secondary infection. Furthermore, hormone-synchronized mice of the same age, gender (female), and mouse strain (C57BL/6J) were used, and the amount of chlamydia in the secondary infection and in the challenge infection after vaccination was identical and applied in the same manner. Thus, the experimental design of the two studies is extremely similar and permits direct comparison of the obtained results. The vaccine-induced protection closely resembles—in its ameliorated course and severity of disease—that of live-infection-induced protection, depicted by body weight and clinical score, and further confirmed by the bacterial load and all others 1 week after the challenge infection measured parameters. Most importantly, the application of adjuvanted 5cVAC derived from *C.tr*. serovar E led to a high degree of cross-serovar protection against *C.tr.* D, E, L2, and even A [26], without eliciting any immunopathology in the recipient mice. These results further support the interpretation of the observations in this reinfection study, i.e., that antigens independent of the variable parts of MOMP (in the right composition, combined with the right adjuvant, and applied by the right route) can be sufficient for protection—information that might be important for many research groups developing a prophylactic vaccine against *C.tr*.

Several reports suggest a role of antibodies (e.g., by participation in opsonophagocytosis or neutralization of bacterial binding to host cells) in protection against secondary chlamydial infections, as demonstrated for *C. muridarum* [40,41] and reviewed by Roan and Starnbach [42]. Furthermore, in our hands, the transfer of specific hyperimmune serum before infection with zoonotic *C. psittaci* improved body weight and survival rate in mice with a hampered immune response (in this case, caused by the absence of complement C3a-receptor). This indicated that infection-induced antibodies can be protective, at least in secondary mouse lung infection with this related chlamydial species [43]. In the present study, we found only minor, most likely irrelevant differences in the IgG responses against all three serovars after i.n. secondary infection with the identical *C.tr.* serovar. This result is in accordance with the observed cross-serovar protection. Activated B cells and chlamydia-specific antibodies might also play a role for proper CD4^+^ priming and the control of bacterial dissemination in the urogenital tract, as has been shown for *C. muridarum* [44,45]. It should also be noted that this does not necessarily mean that the observed, induced chlamydia-specific antibodies are neutralizing the binding of EBs or that the raised antibodies must strongly influence defense and the course of the disease. On the contrary, as recently demonstrated by serum transfer, in our model, circulating antibodies play only a minor role in 5cVAC-induced protection against *C.tr.*, if at all [26]. *Chlamydia-trachomatis*-induced cellular immune responses of CD4^+^, CD8^+^, memory, tissue-resident, regulatory, and other T cell subsets, might be equally or even more important than antibodies for the observed protection after reinfection with the intracellular pathogen. In particular, the CD4^+^ T cell subset has been attributed with taking up an especially important role, e.g., as a central mediator during host defense against chlamydia [46,47].

## 5. Conclusions

In summary, in all nine combinations of the three *C.tr.* serovars used in first and second i.n. infections, all acquired parameters of disease (i.e., weight loss, clinical score, and bacterial load in the lung and levels of IFN-γ, TNF-α, as well as MPO) were similarly diminished after secondary infection, indicating strong cross-serovar protection between *C.tr.* D, E, and L2. Moreover, secondary infection with the identical *C.tr.* serovar induced similar IgG responses against all three serovars. The findings in our screening model strongly suggest that, independent of the unknown protective underlying immune mechanism, in principle, efficient cross-serovar protection could be achieved by a vaccine based on *C.tr.* antigens that does not need to include nonconserved MOMP regions. Such a vaccine could reduce the risk of immuno-evasion by *C.tr.* mutants. The results of this study enhance the significance of other antigen-components of a potential vaccine, such as Pmps, Ctad1, and many more.

## Figures and Tables

**Figure 1 vaccines-09-00871-f001:**
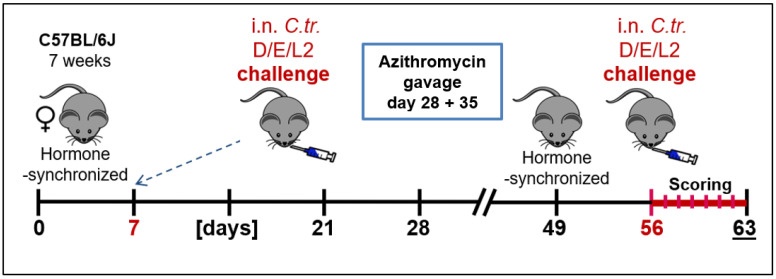
Experimental setup of the *C.tr.* reinfection model with the primary and secondary i.n. infection.

**Figure 2 vaccines-09-00871-f002:**
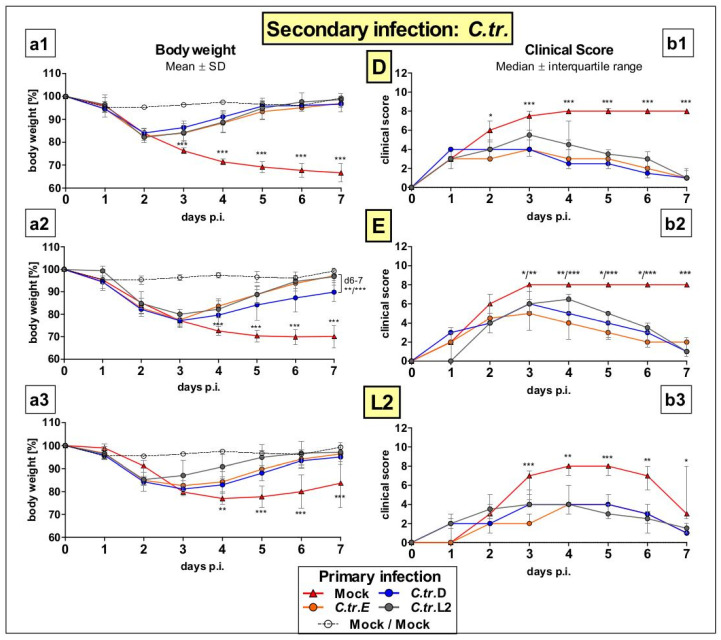
Mice challenged with *C.tr.* serovars D, E or L2 twice, at the time points of primary and secondary infection, show a similar ameliorated loss of body weight and clinical score, almost independent of the combination of serovars used. Following the mouse lung reinfection protocol, i.n. infected animals receiving *C.tr.* genital serovars D, E or L2 or mock material, respectively, were infected again with the *C.tr.* serovars D (upper graphs), E (middle graphs) or L2 (lower graphs) 7 weeks after the primary infection. Mice that were twice infected with mock material served as negative control. Day 0 p.i. in this graph corresponds to day 56 in Figure 1. During the following 7 days, body weight (%; left panels (**a1**–**a3**), mean ± SD) and clinical score (right panels (**b1**–**b3**), median ± interquartile range) were assessed daily. The performed statistical analysis is described in Section 2. Group sizes were *n* = 8–10 with the exception of *n* = 6 for *C.tr.*L2/*C.tr.*L2, and *n* = 4 for Mock/Mock. *–*** indicates statistical significances between the initially mock-infected (red) animal groups and groups previously infected with *C.tr.* D, E or L2 with *p* < 0.05, < 0.01, and < 0.001, respectively. Mice that received mock material twice had a clinical score of 0 (data not shown).

**Figure 3 vaccines-09-00871-f003:**
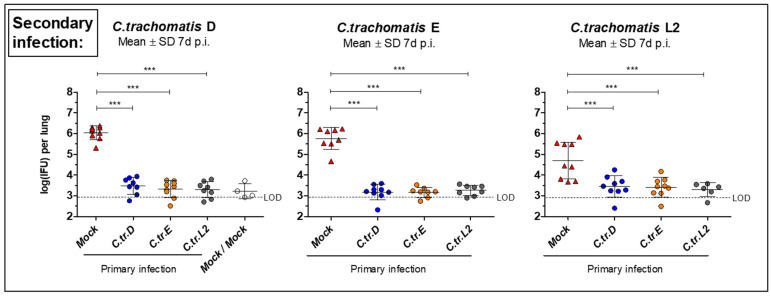
Mice i.n. reinfected with *C.tr.* serovars D, E, and L2 show similar improved bacterial clearance after recovery from a lung infection with either of the three serovars. With *C.tr.* genital serovars D, E or L2 or mock material, pre-challenged animals were infected i.n. again with the *C.tr.* serovars D (left panel), E (middle panel) or L2 (right panel) 7 weeks after the first infection. Mice twice infected with mock material served as negative control. Seven days after the secondary i.n. infection with *C.tr.* serovars D, E or L2, mice were painlessly sacrificed and bacterial load—i.e., the amount of viable, infectious bacteria (left to right panel, mean ± SD of log_10_ (IFU))—was determined. LOD = limit of detection. The performed statistical analysis is described in Section 2. Group sizes were *n* = 8–9 with the exception of *n* = 6 for *C.tr.*L2/*C.tr.*L2, and *n* = 4 for Mock/Mock. *** indicates statistical significances between the previously mock-infected (red) animal groups and groups initially infected with *C.tr.* D, E or L2 with *p* < 0.001.

**Figure 4 vaccines-09-00871-f004:**
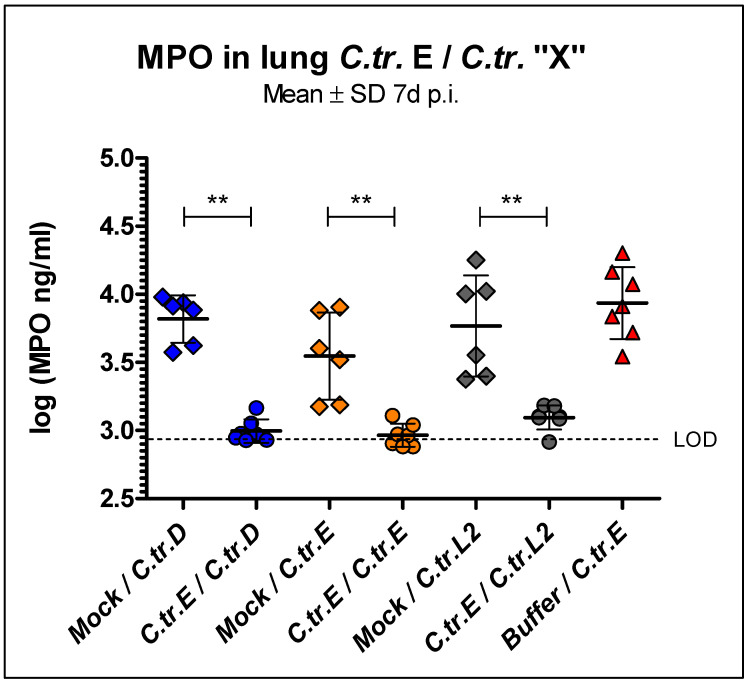
Mice infected i.n. with *C.tr.* serovars D, E, and L2 show lower myeloperoxidase (MPO) concentrations in lung homogenate after overcoming a primary lung infection with *C.tr.* serovar E. Animals previously i.n. infected with *C.tr.* genital serovar E or mock material (*n* = 6–7) were infected a second time with the *C.tr.* serovars D, E or L2 7 weeks after the primary infection. Mice infected twice with mock material (*n* = 4) served as negative control; the MPO-level of all individual control animals was <log(2.95) ng/mL (not shown). Seven days after the second i.n. infection with *C.tr.* serovars D, E or L2, mice were painlessly sacrificed and the level of MPO (mean ± SD of log_10_(IFU)) was determined in lung homogenate. LOD: Limit of detection. The performed statistical analysis is described in Section 2. ** indicates statistical significances between the previously mock-infected animal groups and groups initially infected with *C.tr.* E with *p* < 0.01.

**Figure 5 vaccines-09-00871-f005:**
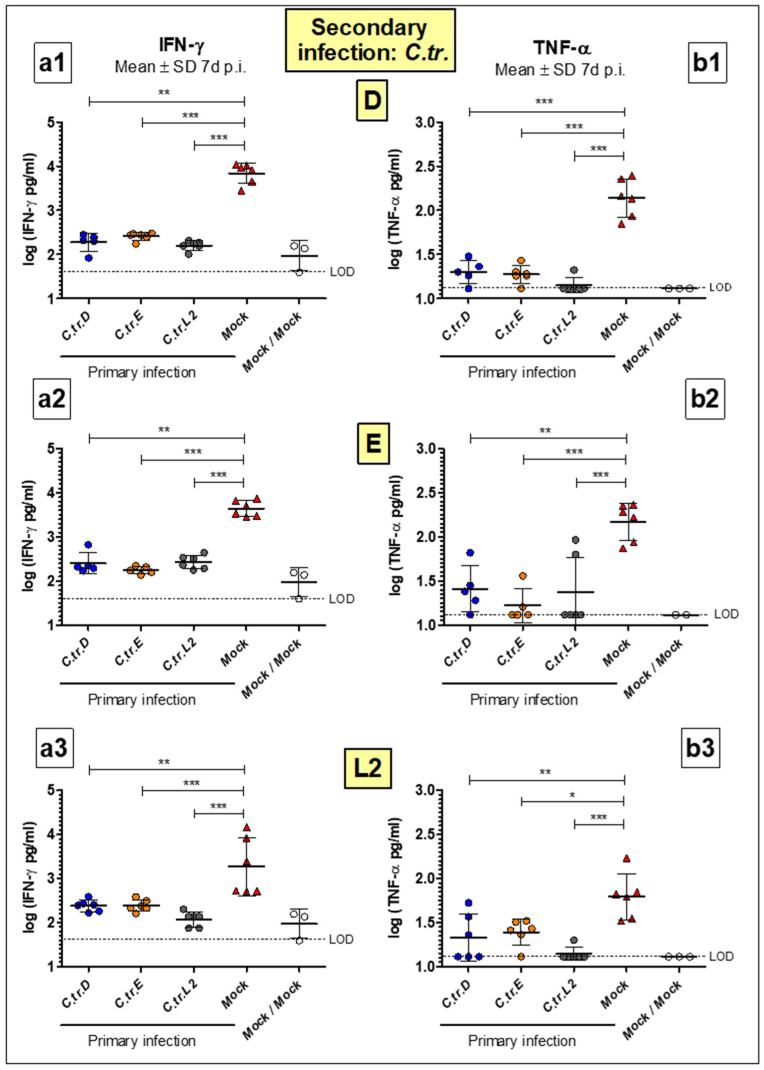
After recovery from a previous lung infection with either of the serovars, mice i.n. reinfected with *C.tr.* serovars D, E, and L2 similarly show lower IFN-γ and TNF-α concentrations in lung homogenate. Animals that were i.n. pre-challenged with *C.tr.* genital serovars D, E or L2 or mock material were infected again with the *C.tr.* serovars D (upper panels), E (middle panels) or L2 (lower panels) 7 weeks after the first infection. Mice twice infected with mock material served as negative control. Seven days after the second i.n. infection with *C.tr.* serovars D, E or L2, mice were painlessly sacrificed and IFN-γ (panels (**a1**–**a3**), mean ± SD of log_10_ (IFU)) and TNF-α (panels (**b1**–**b3**), mean ± SD of log_10_ (IFU)) levels were determined in lung homogenate. LOD: Limit of detection. The performed statistical analysis is described in Section 2. Group sizes were *n* = 5–6 with the exception of *n* = 2–3 for Mock/Mock. *–*** indicates statistical significances between the initially mock-infected (red) animal groups and groups previously infected with *C.tr.* D, E or L2 with *p* < 0.05, <0.01, and <0.001, respectively.

**Figure 6 vaccines-09-00871-f006:**
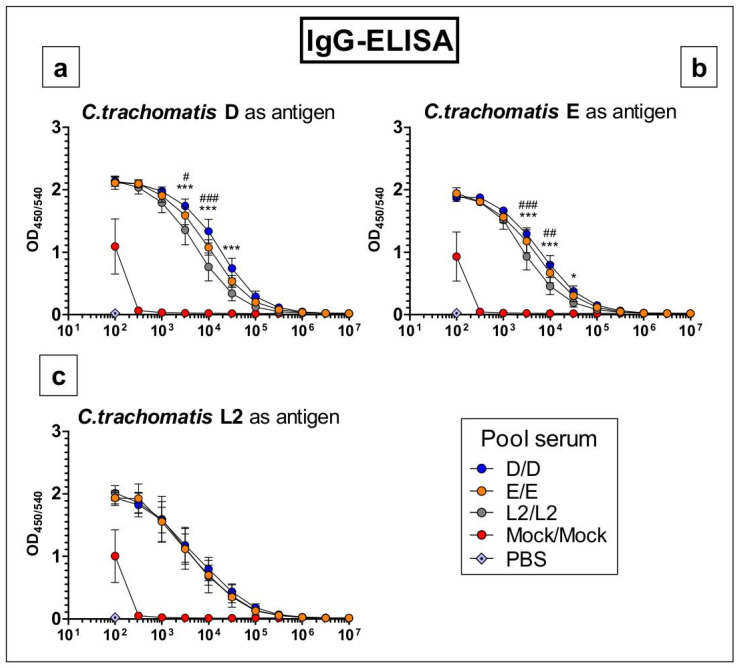
The specific IgG responses against *C.tr.* D (**a**), E (**b**), and L2 (**c**) homogenates determined in mouse serum pools obtained 7 days after homologous primary and secondary infection of mice with the three *C.tr.* serovars are very similar. ELISAs detecting IgG directed against either 2.5 μg/well *C.tr.* D, E or L2 cell homogenate were performed with pool sera obtained from *n* = 7 (D/D), 8 (E/E), 6 (L2/L2), and 4 (Mock) individual mice 7 days after secondary infection with the same serovar. Serial dilution started at 1:100 and continued in 1:3.16 steps (1:316, 1:1000 …) until 1:10 mil. PBS controls were applied at a dilution of 1:100. Experiments were repeated at least once with duplicates measured for each individual dilution. The performed statistical analysis is described in Section 2. *–*** indicates statistical significances between the D/D and L2/L2 pool sera and #–### indicates statistical significances between the E/E and L2/L2 pool sera with *p* < 0.05, <0.01, and <0.001, respectively.

## Data Availability

The raw data supporting the conclusions of this article will be made available by the authors, without undue reservation.

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
