# Peer review of "Chlamydia trachomatis Cross-Serovar Protection during Experimental Lung Reinfection in Mice"

_vaccines, 2021, doi:10.3390/vaccines9080871_

Round 1

Reviewer 1 Report

This is a well designed proof of principle study with clear-cut results well supported by the figures.

There are just some minor comments:

  • Extensive text editing should be done, since a marked “German” construction of sentences was implemented (see 2.6. in particular). Syntax is different between German and English. Some statements are difficult to understand (e.g. first paragraph of results) because of this. Constructions using lots of participles are cumbersome to read. Text editing should include figure legends.
  • Uniform terminology referring to the 1st and 2nd infection should be used. Switching between the terms re-infection, secondary infection and challenge is confusing. This should also include the labelling in figures.
  • The authors clearly state in the introduction that this infection model serves as proof of principle. There should be reminder about this in the discussion/conclusions. Do the authors plan to repeat the study in a more appropriate model? Are the immune responses in the respiratory and urogenital tract comparable?

L62: certain …. Did you mean specific?

L110-115: shorten and condense the statement

L342-347: please reword

L366-368: This statement belongs to “conclusions”.

L386-392: Move the description of methods to the respective section.

L440-451: please reword

Author Response

Reviewer 1:

There are just some minor comments:

  • Extensive text editing should be done, since a marked “German” construction of sentences was implemented (see 2.6. in particular). Syntax is different between German and English. Some statements are difficult to understand (e.g. first paragraph of results) because of this. Constructions using lots of participles are cumbersome to read. Text editing should include figure legends.

We rephrased many parts of the manuscript for easier understanding.
Furthermore, we used the MDPI English Editing Service for optimization of the revised manuscript.

  • Uniform terminology referring to the 1st and 2nd infection should be used. Switching between the terms re-infection, secondary infection and challenge is confusing. This should also include the labelling in figures.

We adapted the terminology for clarification. The model itself is described as the “reinfection model”, and “reinfection” in the text is only being used if applicable.

The two timepoints of infection are now uniformly described as “primary” and “secondary” infection in both figures and text.

The term “challenge” has been replaced as far as possible and is now mainly used in conjunction with the previous vaccination work or other models.

  • The authors clearly state in the introduction that this infection model serves as proof of principle. There should be reminder about this in the discussion/conclusions. Do the authors plan to repeat the study in a more appropriate model? Are the immune responses in the respiratory and urogenital tract comparable?

In the modified version of our manuscript, we mention this point (proof of principle) now three times, in Introduction, Results and Discussion. Furthermore, in Conclusions we use as a similar expression “the findings in our screening model…”. Presently, we are not planning to perform an additional (extensive) reinfection study in the urogenital setting. Instead, we are pursuing vaccine development  in the urogenital mouse model (with non-MOMP antigens). 

L62: certain …. Did you mean specific?

Yes, we replaced the word.

L110-115: shorten and condense the statement

We rephrased and split the statement into two shorter sentences for better readability.

L342-347: please reword

Reworded the figure legend for better readability.

L366-368: This statement belongs to “conclusions”.

We removed the statement at lines 366-368, modified and inserted it in the conclusion section 5.

L386-392: Move the description of methods to the respective section.

We removed the description and (after removing redundant parts) included it in the section 2.5 of Material and Methods.

L440-451: please reword

Rephrased the entire paragraph for clarification.

Reviewer 2 Report

Overall, a well written manuscript and designed study.  The one question this reviewer had that stood out was the authors didn't show the antibody titers post vaccination, rather it was done 7 days post challenge.  Could the authors address this concern?

Author Response

Reviewer 2:

Overall, a well written manuscript and designed study.  The one question this reviewer had that stood out was the authors didn't show the antibody titers post vaccination, rather it was done 7 days post challenge.  Could the authors address this concern?

We explain now in the results section „We analyzed only pool sera obtained after reinfection with the identical serovar because orientating preliminary experiments had indicated higher sensitivity (data not shown).“ It seems highly unlikely that differences in serovar specificity should only be visible before reinfection. Moreover, according to the experimental protocol the German authorities had agreed to, the animals were sacrificed 7 days after secondary infection. Thus, for practical reasons, we were limited to material obtained at that time-point.